# Single-Stage Deep Brain Stimulator Placement for Movement Disorders: A Case Series

**DOI:** 10.3390/brainsci11050592

**Published:** 2021-05-03

**Authors:** Arrin Brooks, Alastair T. Hoyt

**Affiliations:** 1Department of Biomedical Research, Marshall University, Joan C Edwards School of Medicine, Huntington, WV 25701, USA; 2Department of Neurosurgery, Joan C Edwards School of Medicine, Huntington, WV 25701, USA; hoyta@marshall.edu

**Keywords:** deep brain stimulation, interoperative neurophysiology, movement disorders, stereotactic surgery, surgical technique

## Abstract

With more than two decades of experience and thousands of patients treated worldwide, deep brain stimulation (DBS) has established itself as an efficacious and common surgical treatment for movement disorders. However, a substantial majority of patients in the United States still undergo multiple, “staged” surgeries to implant a DBS system. Despite several reports suggesting no significant difference in complications or efficacy between staged and non-staged approaches, the continued use of staging implies surgeons harbor continued reservations about placing all portions of a system during the index procedure. In an effort to eliminate multiple surgeries and simplify patient care, DBS implantations at our institution have been routinely performed in a single surgery over the past four years. Patients who underwent placement of new DBS systems at our institution from January 2016 to June 2019 were identified and their records were reviewed. Revision surgeries were excluded. Total operative time, length of stay and rates of surgical site infections, lead fracture or migration, and other complications were evaluated. This series expands the body of evidence suggesting placement of a complete DBS system during a single procedure appears to be an efficacious and well-tolerated option.

## 1. Introduction

Deep brain stimulation (DBS) therapy has become a mainstay for movement disorders including essential tremor, Parkinson’s disease (PD), and dystonia [1]. More than 160,000 patients have been implanted worldwide and more than 12,000 new patients receive DBS annually [2]. Despite decades of experience, patients are frequently offered implantation via multiple, “staged” surgeries. In fact, a 2013 international survey of DBS procedural steps revealed that DBS implantations occurring in the United States were more likely to be staged than those at European centers [3]. In America, 65% percent of DBS surgeries were staged in 2 days, 22% staged in 3 days and only 13% were performed in a single stage.

Reasons to separate DBS implantation into multiple surgeries vary, including concerns for increased risk of infection, poor stimulation efficacy, and postoperative confusion leading to increased length of stay, particularly when intraoperative microelectrode (MER) recording and neurophysiological testing are used [4,5]. In an effort to simplify care and reduce the burden of multiple surgeries on patients, a single-stage approach has been used routinely in the senior author’s practice since his arrival at our institution. We hypothesize that single-stage DBS placement surgery has a similar risk profile, similar or shorter length of hospital stay, and similar or reduced rate of reoperations compared to the reported outcomes of surgeries performed in multiple stages.

## 2. Materials and Methods

### 2.1. Patient Selection

After institutional review board approval (IRB# 1447599-2), a retrospective review of clinical data from our institution was undertaken for a cohort of patients who underwent DBS implantation from January 2016 to June 2019. The inclusion criteria included all patients receiving initial placement of DBS system or second unilateral placement (no previous electrode in that hemisphere). Revision surgeries and routine implanted pulse generator (IPG) exchanges were excluded.

### 2.2. Surgical Technique

All patients underwent preoperative neurology evaluation, neuropsychological testing, and on/off testing in the case of Parkinson’s disease. A stereotactic protocol imaging study was obtained prior to the day of surgery. On the day of surgery, a Leksell Model G Frame (Elekta, Stockholm, Sweden) was fitted under conscious sedation and a registration CT scan was performed, followed by computer-aided surgical planning. Depending on the stimulation target and patient, either “awake” placement with MER and stimulation testing or “asleep” placement under general anesthesia was performed. Intraoperative IPG’s and extension leads were routinely placed prior to exiting the operating theater. All patients underwent a postoperative CT scan of the head to assess for intracranial complications. As our institution does not allow scheduled nursing assessments more frequently than every four hours in general hospital beds, patients were observed in the intensive care unit for neurological monitoring and blood pressure control. Initial programming of the stimulation system typically occurred two to four weeks after surgery. Patients who did not obtain adequate symptom control, as determined by the patient and the treating neurologist, were offered DBS revision surgery.

### 2.3. Literature Review

Similar reports were identified by a structured search of the PubMed database and were summarized.

## 3. Results

A total of 73 patients that met inclusion criteria were identified. The average age was 65.7 years (range 41–80). Of these patients, 20 were female, and 53 were male. Mean operative time was 3:50 (range 2:30–5:44) for bilateral placements and 2:58 (range 1:59–4:14) for unilateral placements (Figure 1). The mean length of stay was 1.2 days (range 1–3), with 86.3 percent of patients discharged on the first postoperative day. The average follow-up after surgery was 23.3 months (range 2.3–50.9). Of the 73 total patients: 46 were treated for PD, 26 for essential tremor (ET), and one for dystonia. ET was treated with thalamic stimulation and dystonia with pallidal stimulation, while PD was treated with either pallidal or subthalamic stimulation (Figure 2). There were four patients (5.5%) that underwent “asleep” DBS without MER (Figure 2). The average number of microelectrode recording passes per electrode placed was 1.55.

There were fifty-six patients (77%) who underwent bilateral electrode placement and 15 patients (20%) who underwent unilateral electrode placement. Of these, two patients (3%) were scheduled to undergo bilateral electrode placement but surgery was halted after placement of the first electrode, one for patient fatigue and one for equipment failure. Both had a second electrode placed in a later procedure.

Table 1 demonstrates the post-operative complications observed. All patients recovered from their complications, with no mortalities or permanent neurological injuries. Complications were suffered within 90 days of surgery by six patients (8.2%), while five (6.8%) suffered a complication more than 90 days after surgery. The most frequent complication was wound infection (5.5%), which was treated aggressively with the removal of the affected portions of the stimulation system and antibiotic treatment. There were six patients (8.2%) readmitted within 90 days of surgery (Table 2). Of these six, four required surgery for treatment of a complication, including two infections, one lead fracture, and one generator site hematoma. A single patient was readmitted with poorly defined symptoms which resolved after treatment of pain with nonsteroidal anti-inflammatories. Another patient had bradycardia and hypotension due to cardiac disease not deemed related to surgery or stimulation. In total, 11 patients (15.1%) underwent additional surgery during the follow-up period, nine (12.3%) for a complication, and the remainder for poor stimulation effect (See Figure 3).

There were two patients (2.7%) that underwent a revision for poor stimulation effect. The first was a 66-year-old man with a diagnosis of PD. Pre-operative on/off testing resulted had revealed significant a reduction of UPDRS motor scores from 55 to 27. He underwent bilateral pallidal stimulator placement but had little improvement in symptoms with stimulation. MRI imaging confirmed lead placement in the desired target bilaterally. He subsequently underwent revision of the electrodes from the pallidum to the subthalamic nucleus, again with only marginal improvement in symptoms.

The second was a 67-year-old man with bilateral upper extremity tremor diagnosed as essential tremor. He underwent bilateral “asleep” thalamic DBS due to a history of severe sleep apnea. Substantial improvement was noted in tremor for about 6 weeks after initial programming, and then tremor appeared to return. Despite multiple programming attempts, the tremor worsened and he reported persistent headaches. He did not respond to a trial of levodopa. He ultimately underwent bilateral “awake” lead revision with MER, although he had relatively little tremor to test intraoperatively. He again experienced no improvement in tremor despite multiple rounds of programming and ultimately elected to have the DBS removed due to persistent headaches.

## 4. Discussion

While DBS is a well-established therapy for movement disorders, there is considerable variation in methods of system placement. Attempts to determine the best methods are hindered by differences in terminology, inconsistent reporting of complications and outcomes, and the individual practices of surgeons. It is clear, however, that the vast majority of patients in the United States have been made to undergo multiple staged procedures [3]. Little data exists to suggest the practice of staging is superior. Arguments for staging have included improved electrode accuracy, reduced operative time, and concerns for increased post-operative confusion or other complications [4,5].

Some authors have referred to “staging” as individually placing only one of two planned intracranial electrodes at a time, and studies have compared simultaneous bilateral electrode placement with implantation of each hemisphere in two separate surgeries [5,6]. Peng, et al. concluded that there was no significant difference in lead accuracy between unilateral, simultaneous bilateral, and staged bilateral electrode placement [5]. Petraglia, et al. reported a higher rate of revision within 90 days with separate unilateral electrode placement than in simultaneous bilateral electrode placement but found no significant difference between these groups in rates of complication or costs [6]. Neither author reported when an IPG was implanted (Table 3).

Similarly, there have been several published comparisons of “awake” surgery including MER and intraoperative neurophysiological testing with “asleep” surgery under general anesthesia. A clearly superior technique has not been established [7]. Studies have shown similar outcomes when comparing intraoperative CT (iCT)-guided sub-thalamic nucleus (STN)- or Globus Pallidus (Gpi)-DBS lead implantation to MER-guided DBS [8], as well as when comparing iMRI-guided GPi-DBS to MER-guided DBS [9], with some motor outcomes slightly favoring asleep DBS. Mirzadeh, et al. suggested stereotactic accuracy was better under general anesthesia without MER, compared to the group receiving MER intraoperatively [10,11]. Opposingly, in a retrospective analysis of awake versus asleep DBS for STN stimulation in PD, Blasberg, et al. found motor function improved faster following awake surgery and axial subitems were worse in the asleep surgery group [12]. They concluded awake placement was advantageous over asleep placement in suitable patients. Still, others have reported no significant differences in complications, length of stay, and readmissions between awake and asleep DBS procedures [13].

Our observations indicate that MER and “awake” stimulation testing result in superior electrode placement, particularly in small targets like the STN. As a result, placement under general anesthesia was reserved for patients with significant respiratory comorbidities. In this series, the need for revision due to poor effect was used as a surrogate for stimulation efficacy. There were two patients (2.7%) that underwent a revision for poor effect. A patient with a diagnosis of PD did not have a good response even after revision of the electrodes from the pallidum to the subthalamic nucleus, suggesting that it was not poor placement but the patient’s disease that resulted in lackluster symptom control. A second patient with ET had a transient response to thalamic stimulation and no improvement after revision of the electrode, again suggesting a flaw in patient selection and not in electrode placement.

An increased operative time is a logical consequence of single-stage surgery. While we did observe a longer operative time than is reported in some literature, we did not observe any serious complications intra- or peri-operatively related to this increased time under sedation. On one occasion, a bilateral implantation surgery was halted after placement of a single due to patient somnolence. The patient recovered normally and received the second electrode in a later procedure. One additional patient had bradycardia and hypotension due to cardiac disease not deemed related to the length of surgery. In both instances, placement of the IPG in a second procedure would not have prevented the event. With an average length of stay of 1.2 days, and 86% of patients returning home on the day after surgery, this increased operative time did not appear to result in significantly longer lengths of stay (Table 3).

Further, while difficult to quantify, the single-stage approach reduces each patient’s total time commitment to DBS placement by avoiding procedures on multiple days. In a similar effort to streamline the implantation process, Van Horne, et al. [14] performed cranial access, IPG placement, and extension lead insertion during an initial procedure and stimulating lead placement using MER in the second stage. An argued benefit was that an entire system was implanted and ready to program within five days. We assert that a nonstaged approach provides similar advantages in a single day.

Variation in how complications are reported frustrates direct comparison to other approaches, but the presented results are similar to published accounts [6,13,15,16,17,18,19,20,21,22,23,24,25] (See Table 3). In a study evaluating factors associated with postoperative confusion and prolonged hospital stay following DBS surgery for PD, Abboud, et al. determined that surgical factors such as implantation laterality, surgical staging, or the number of MER passes did not influence immediate outcomes [26]. A European study showed a nonsignificant decrease in infection rate in single stage (entire DBS system placed in one operative session) DBS surgery compared to that in staged DBS surgery (IPG placed in subsequent surgery) [27]. In a retrospective analysis of “asleep” DBS, Chen et al. found that staging did not significantly affect hardware-related complications such as infection, erosion, impedance, or lead malposition [13]. They also observed a reduction in hemorrhage or seizures with staging. In another study categorizing staging in the same manner described in this study, surgical site infections were shown to be similar in 228 staged and 17 non-staged implantations (6.6% to 5.9%) [23]. Their results also showed similar rates of wound dehiscence and post-op seroma between staged and non-staged groups (Table 3).

Perhaps one of the most important factors to many surgeon’s decision to perform staged DBS implantation is financial. Given the complexities of medical economics in the United States, it is very difficult to perform an economic evaluation of cost-effectiveness for DBS as a whole, let alone determining the relative cost-effectiveness of single-stage procedures [28]. The current payment structure reimburses hospitals more based on Medicare Severity—Diagnosis Related Group (MS-DRG) codes for staged procedures [29]. For example, staged placement of bilateral subthalamic electrodes in one procedure (MS-DRG 027) with later placement of an IPG (MS-DRG 042) results in 5.7% greater relative weight for reimbursement than nonstaged placement (MS-DRG 024). This payment structure incentivizes hospitals to mandate surgeons to perform staged procedures and unfortunately may result in patients having unnecessarily convoluted care.

While admittedly a small sample, this report constitutes one of the larger series of frame-based bilateral electrode placement with intraoperative MER and neurophysiological testing accompanied by IPG placement in the United States. Direct comparison to staged approaches is limited by the lack of a control group of staged surgeries at the same facility by the same surgeon. Anecdotally, the authors have found avoiding multiple procedures to be attractive to patients. It helps to simplify scheduling and to reduce transportation needs. These factors are particularly attractive when patients may travel several hours to reach the hospital, such as rural settings. While no formal assessment has been performed, patients frequently report that they prefer a single-stage approach as the longer procedure does not necessitate a longer hospital stay. Performing a systematic assessment of patient’s attitudes toward a longer, single procedure versus several shorter procedures would help support this assertion.

## 5. Conclusions

This series expands the body of evidence suggesting placement of a complete DBS system during a single procedure, including bilateral electrodes with MER, which appears to be an efficacious and well-tolerated implantation option. Complication rates, length of stay, and readmission rates appear comparable with published reports of staged procedures while the patient avoids multiple procedures.

## Figures and Tables

**Figure 1 brainsci-11-00592-f001:**
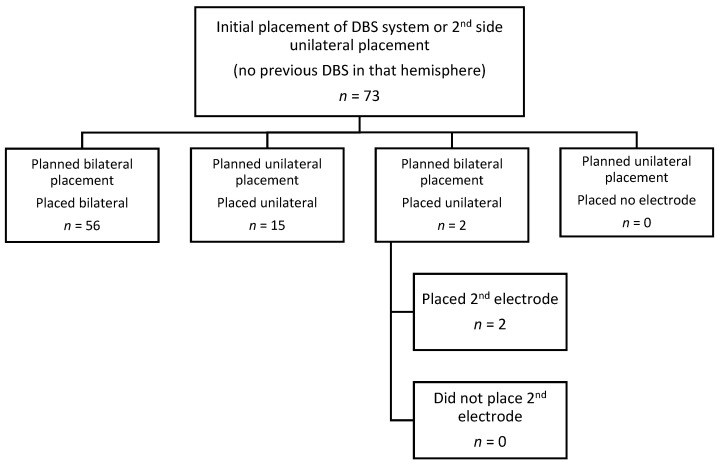
Unilateral versus bilateral placement in included patients.

**Figure 2 brainsci-11-00592-f002:**
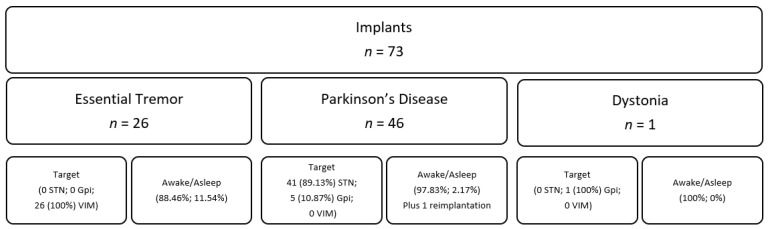
Conditions treated and brain targets in included patients.

**Figure 3 brainsci-11-00592-f003:**
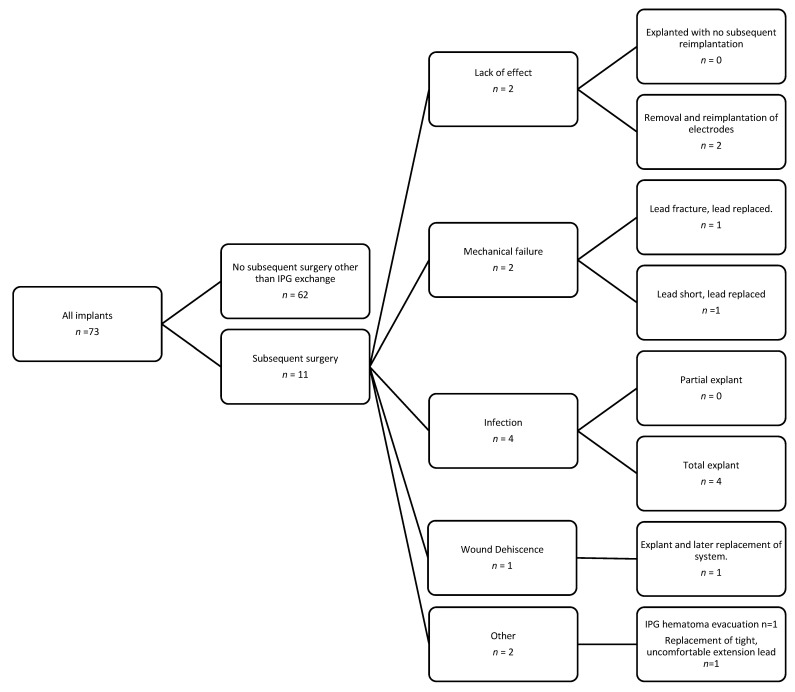
Flow chart of outcomes.

**Table 1 brainsci-11-00592-t001:** Complications suffered by patients within 90 days of surgery and at 90 days or longer after surgery.

	Within 90 Days		>90 Days		Total (as of the Last Follow-Up)	
	Number	Percent (of All Patients)	Number	Percent (of All Patients)	Number	Percent (of All Patients)
Infection	2	2.7%	2	2.7%	4	5.5%
Lead Fracture	1	1.4%	1	1.4%	2	2.7%
Wound Dehiscence	0	0%	2	2.7%	2	2.7%
Hypotension	1	1.4%	0	0%	1	1.4%
IPG Site Hematoma	1	1.4%	0	0%	1	1.4%
Total	6	8.2%	5	6.8%	11	15.1%

**Table 2 brainsci-11-00592-t002:** Readmissions within 90 days of surgery.

	Number	Percent (of All Patients Included)
Infection	2	2.7%
Lead Fracture	1	1.4%
Hypotension	1	1.4%
IPG Site Hematoma	1	1.4%
Pain	1	1.4%
Total	6	8.2%

**Table 3 brainsci-11-00592-t003:** Complications in other published works.

	No. of Patients (Electrodes)	Staging	Infection	Hardware Complication	Wound Dehiscence/Erosion	Hemorrhage	Seizures	Loss of Efficacy	Other
Abode-Iyamah, et al. (2019)	242 (464)	S: 228 N: 17	S 15 (6.6%) N 1 (5.9%)	NR	S: 9 (4.0%) N: 0 (0%)	NR	NR	NR	Postoperative Seroma S: 1 (0.4%) N: 0 (0%)
Chen, et al. (2017)	284 (490)	S: 200 N: 84	S: 3 (1.5%) N: 0 (0%)	High impedance S: 1 (0.5%) N: 1 (1.2%)	S: 1 (0.3) N: 0 (0%)	S: 3 (1.5%) N: 1 (1.2%)	S: 2 (1%) N: 2 (2.4%)	NR	
Fenoy (2014)	728 (1333)	Transitioned from N to S, data not segregated	23 (3.2%)	Lead malposition 9 (1.2%) Lead migration 4 (0.5%) High impedance 4 (0.5%) Fracture 10 (1.4%)	4 (0.5%)	Symptomatic 8 (1.1%) Asymptomatic IVH 25 (3.4%) ICH 4 (0.5%)	5 (0.7%)	29 (4%)	IPG flipped, malpositioned or discomfort 8 (1.1%)
Petraglia (2016)	713 (1426)	Sim Bilat: 556 Staged Bilat: 157 IPG placement timing NR	Sim Bilat: 24 (4.3%) Staged Bilat: 11 (7%)	Sim Bilat: 3 (0.5%) Staged Bilat: 0 (0%)	NR	Sim Bilat: 16 (2.9%) Staged Bilat: 4 (2.5%)	NR	NR	Lead Revision Sim Bilat: 18 (3.2%) Staged Bilat: 20 (12.7%) Generator Revision Sim Bilat: 17 (3.1%) Staged Bilat: 6 (3.8%)
Doshi (2010)	153 (298)	All staged	6 (3.9%)	Lead malposition 4 (2.6%) Lead migration 0 (0%) IPG malfunction 2 (1.3%) Fracture 0 (0%)	1 (0.7%)	2 (1.3%)	NR	2 (1.3%)	
Voges, et al. (2006)	262 (472) 180 (352) assessed for long-term complications	S: 194 (74.1%) N: 64 (24.4%) data not segregated	15 (5.7%)	Electrode damage/fracture 4 (2.2%) Local discomfort 12 (6.7%) Electrode migration 5 (2.8%) Connector displacement 1 (0.6%)	1 (0.6%)	1 (0.4%)	0 (0%)	NR	IPG implantation site hematoma 3 (1.25) Seroma at IPG site 2 (1.1%)
Seijo, et al. (2011)	130 (252)	All staged	2 (1.5%)	Lead fracture 1 (0.8%)	NR	9 (6.9%)	13 (10%)	NR	CSF leak 1 (0.8%)
Tolleson, et al. (2014)	447 (823)	All staged	26 (5.8%)	Lead migration 2 (0.5%) Lead and IPG malfunction 2 (0.5%)	Infected group 8 (1.8%) Noninfected group 9 (2%)	2 (0.5%)	1 (0.2%)	NR	Pain along apparatus 4 (0.9%)
Kochanski, et al. (2019)	178 (270)	Both Staged and Nonstaged procedures include; not segregated	3 (1.7%)	Malpositioned lead 0 (0%)	NR	3 (1.7%)	2 (1.1%)	NR	
Falowski, et al. (2015)	432 (606)	S: 326 (475) N: 106 (131)	S: 11 (3.4%) N: 4 (3.8%)	Lead fractures S 7 (2.1%) N 7 (6.6%) High impedance S 5 (1.5%) N NR Lead migration S 1 (0.3%) N NR	S: 3 (0.9%) N: NR	S: 12 (3.7%) N: 8 (7.5%)	NR	S: 17 (5.2%) N: NR	Extension lead coiling S: 2 (0.6%) N: NR Seroma S: 1 (0.3%) N: NR
Morishita, et al. (2017)	132 (138)	All staged	17 (12.9%)	Fracture 7 (5.1%) Lead migration 16 (12.1%)	NR	Symptomatic ICH 5 (3.8%) Asymptomatic ICH 4 (3%)	9 (6.8%)	NR	Air embolus 2 (1.5%)

Total number (% of event by number of patients). S = Staged (IPG implanted in subsequent surgery); N = Nonstaged (DBS system including IPG implanted in one operative session); Sim Bilat = Simultaneous Bilateral (DBS leads implanted in each hemisphere in one surgery); Staged Bilat = Staged Bilateral (each hemisphere implanted in staged surgeries); NR = Not Reported; ICH = intracranial hemorrhage; IVH = intraventricular hemorrhage.

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
