# Peer review of "Single-Stage Deep Brain Stimulator Placement for Movement Disorders: A Case Series"

_brainsci, 2021, doi:10.3390/brainsci11050592_

Round 1

Reviewer 1 Report

The authors report on a series of 73 DBS surgeries for Parkinson disease (n=46) or essential tremor (n=26) performed between 2016 and 2019 at their center. Most of the patients (77%) received bilateral implantation of electrodes and placement of the impulse generator during a single surgery, while 20% underwent unilateral surgery and placement of the impulse generator during a single surgery (single-stage). Most of the operations (95%) were done awake. Mean operative time was 2.58 and 3.50 hours for unilateral und bilateral surgery, respectively. Mean length of hospital stay was just 1.2 days.

The number of complications was about 8% within 90 days and about 7% after more than 90 days after operation. On the whole, this number of reported complications compares favorably with those reported by other groups.

The number of patients with requirement for additional surgery is not entirely clear: n=4 removal of the stimulation systemd due to infection (line 89), n=4 additional surgery for complications including infection (line 91) and n=9 additional surgery for complications (line 95) - possibly related to different time-frames? This should be clarified, and the reasons for reoperation should be stated. It should be mentioned how many complications resolved and how many were permanent.

It is not reported how these data compare to those from patients undergoing staged surgery at their institution before 2016. Although standards of care might have changed over the years, this comparison might provide further suggestions that non-staged procedures can be advantageous or non-inferior at least.

The authors compare their data to other series from the literature. It is not reported how the cited studies have been found or chosen. Some of the studies seem to suggest more complications following staged surgery (wound dehiscence: Abodelyamah; infection: Chen, Petraglia, Pepper; lead revision: Petraglia), but some studies reported somewhat lower rates of lead fractures and / or hemorrhages in staged surgery (Falovski, Chen). On the other hand, treatment had not been randomized in these studies, group sizes were different, many studies were monocentric, only some studies reported complication rates separately for staged and non-staged procedures, complications numbers per group were small, and possible sources of bias were not explored.

Limitations of the study are its retrospective, monocentric, non-randomized design, the rather small number of patients and the lack of a control group.

Furthermore, it would be very helpful to include the patients‘ perspective. Do most patients welcome the lower number of operations? Or do they fear longer surgery time and feel safer with planned readmission for resurgery?   

Author Response

Comment/Suggestion:

The number of patients with requirement for additional surgery is not entirely clear: n=4 removal of the stimulation system due to infection (line 89), n=4 additional surgery for complications including infection (line 91) and n=9 additional surgery for complications (line 95) - possibly related to different time-frames? This should be clarified, and the reasons for reoperation should be stated. It should be mentioned how many complications resolved and how many were permanent.

Response:

The authors agree that the indications for additional surgery was confusing in the text.  We have added the following figure for clarification.

Comment/Suggestion:

It is not reported how these data compare to those from patients undergoing staged surgery at their institution before 2016. Although standards of care might have changed over the years, this comparison might provide further suggestions that non-staged procedures can be advantageous or non-inferior at least.

Response:

We agree that a matched group of staged surgeries at the same institution would result in a more meaningful report than a single provider case series.  However, the senior author has been performing single stage surgeries since his arrival at this institution.  As such, any historical comparison would be made against a different surgeon.  Reliable records of complications, operative time, etc are difficult to identify due to a change in the medical records system at our hospital and poor prior documentation.  It was felt that there would be too many confounders for a reasonable comparison.

Comment/Suggestion:

The authors compare their data to other series from the literature. It is not reported how the cited studies have been found or chosen. Some of the studies seem to suggest more complications following staged surgery (wound dehiscence: Abodelyamah; infection: Chen, Petraglia, Pepper; lead revision: Petraglia), but some studies reported somewhat lower rates of lead fractures and / or hemorrhages in staged surgery (Falovski, Chen). On the other hand, treatment had not been randomized in these studies, group sizes were different, many studies were monocentric, only some studies reported complication rates separately for staged and non-staged procedures, complications numbers per group were small, and possible sources of bias were not explored.

Response:

The authors sought to identify the currently reported patient outcomes following single-stage and multistage DBS implantation surgery. Similar reports were identified by a structured search of the PubMed database and are summarized in Table 3.  As there is currently no published work directly comparing surgical parameters or complication rates between staged and nonstaged DBS implantation in the manner we describe here, the authors conducted a search of Pubmed using combination of the following terms/phrases: DBS, deep brain stimulator, staged, nonstaged, single stage, complications, outcomes. Publications before year 2000 were excluded. Titles describing DBS implantation surgical technique and/or factors affecting intra- and post-op outcomes were selected for review. Only articles that described how surgeries were staged (timing of all electrode and generator placement) were included in the final comparison.

Comment/Suggestion:

Furthermore, it would be very helpful to include the patients‘ perspective. Do most patients welcome the lower number of operations? Or do they fear longer surgery time and feel safer with planned readmission for resurgery?   

Response:

While no formal assessment has been performed, patients frequently report that they prefer a single-stage approach as the longer procedure does not necessitate a longer hospital stay.   The authors feel this would be an excellent topic for further study.  Please see the revisions in the text.

Reviewer 2 Report

The authors present a single-center series of patients submitted to DBS for Parkinson's disease or dystonia. Surgical procedures were performed in a single operating session: after electrode placement, also IPG implants were performed. 

I absolutely agree with the authors about the value of non-staged surgical procedure, both in terms of patients management, in length of stay, and about complication rate. In our center we perform full DBS placement in a single surgical section.

I have only some issues:

-one is related to post-operative imaging: the authors stated that "all patients underwent a postoperative CT scan of the head". It seems that the CT scan was performed at the end of the full procedure (in other words, after the IPG implant). However, I think that performing CT scan after electrode placement, BEFORE IPG implantation, could address and avoid possible re-du surgery in case of malpositioning, before the surgical part performed under general anesthesia. Authors should better clarify this point

-The second issue is related to ICU management: is really necessary? Because the potentially most dangerous surgical step is usually performed under asleep anesthesia, and the step under general anesthesia (so requiring some post-operative monitoring) is more limited. In my experiences, only were few patients, except for those presenting some relevant comorbidities, need an ICU stay. Does also factors, as for example economical policies, have an impact on some of these choices? The authors clearly affirm that reimbursement policies could have an impact, in general, on the decision or tradition to perform staged versus non-staged full DBS placement.

-The last issue is related to the choice of full stimulation system removal in case of wound infection. This happens also in the case of subclavian wound infection, so very far from the intracranial electrodes?

There are different experiences, in literature, about the attempt (often successful) to preserve at least the intracranial electrode (see i.e. Levi V et al, Antibiotic Impregnated Catheter Coating Technique for Deep Brain Stimulation Hardware Infection: An Effective Method to Avoid Intracranial Lead Removal. Oper Neurosurg (Hagerstown). 2020 Mar 1;18(3):246-253. doi: 10.1093/ons/opz118)

Author Response

Comment/Suggestion:

One is related to post-operative imaging: the authors stated that "all patients underwent a postoperative CT scan of the head". It seems that the CT scan was performed at the end of the full procedure (in other words, after the IPG implant). However, I think that performing CT scan after electrode placement, BEFORE IPG implantation, could address and avoid possible re-du surgery in case of malpositioning, before the surgical part performed under general anesthesia. Authors should better clarify this point

Responses:

CT imaging was performed routinely after the general anesthesia portion of surgery to ensure that there was not evidence of intracranial hemorrhage or excessive pneumocephalus.  These images are typically not used to confirm electrode placement as intraoperative MER and stimulation findings confirm appropriate placement.  However, our institution has recently acquired an intra-op imaging capability.  We now routinely perform registration scans in the operating room and perform post-operative stereotactic confirmation scans on all “asleep” DBS placements.  Please see revisions for clarification in the text.

Comment/Suggestion:

The second issue is related to ICU management: is really necessary? Because the potentially most dangerous surgical step is usually performed under asleep anesthesia, and the step under general anesthesia (so requiring some post-operative monitoring) is more limited. In my experiences, only were few patients, except for those presenting some relevant comorbidities, need an ICU stay. Does also factors, as for example economical policies, have an impact on some of these choices? The authors clearly affirm that reimbursement policies could have an impact, in general, on the decision or tradition to perform staged versus non-staged full DBS placement.

Responses:

Institutional rules at our hospital require that neurological assessments and vital signs cannot be routinely performed by nursing staff more frequently than every four hours and IV medications cannot be given for acute blood pressure control outside of the ICU setting.  The authors heartily agree that nearly all patients do not require ICU care, and that such care drives up the costs, but also feel that patients should be observed more closely than every 4 hours after intracranial lead placement.  A “step down” environment would be ideal, but is unfortunately not available at our center.  Please see revisions for clarification in the text.

Comment/Suggestion:

The last issue is related to the choice of full stimulation system removal in case of wound infection. This happens also in the case of subclavian wound infection, so very far from the intracranial electrodes?

Responses:

The authors agree that treatment of isolated IPG site infections can sometime be treated while salvaging the remainder of the system.  However, two of the four infections presented here manifested with findings at the cranial sites.  In one of the remaining infections, there was evidence of purulence at the cranial connection to the extension lead during removal of the extension lead.  In the final incidence, an attempt was made to salvage a portion of the system, but there were recurrent infectious findings which prompted subsequent removal of the entire system.  The authors omitted these details as they appeared to make the text overly cumbersome.

Round 2

Reviewer 2 Report

The authors addressed all my questions, and I think that the manuscript is now suitable for publication